# DNDesign: Denoising is All You Need for Protein Inverse Folding

## Abstract

Based on the central dogma that protein structure determines its functionality, an important approach for protein sequence design is to identify promising sequences that fold into pre-designed structures based on domain knowledge. Numerous studies have introduced deep generative model-based inverse-folding, which utilizes various generative models to translate fixed backbones to corresponding sequences. In this work, we reveal that denoising training enables models to deeply capture the protein energy landscape, which previous models do not fully leverage. Based on this, we propose a novel Denoising-enhanced protein fixed backbone design (DNDesign), which combines conventional inverse-folding networks with a novel plug-in module, which learns the physical understanding via denoising training and transfers the knowledge to the entire network. Through extensive experiments, we demonstrate that DNDesign can easily be integrated into state-of-the-art models and improve performance in multiple modes, including auto-regressive, non-auto-regressive, and scaled-up scenarios. Furthermore, we introduce a fixed backbone conservation analysis based on potential energy changes, which confirms that DNDesign ensures more energetically favorable inverse-folding.

## 1   Introduction

Proteins play a crucial role in various biological processes, serving as the primary components of cellular machinery and exhibiting specific functionalities based on their three-dimensional structures. Therefore, designing unique protein sequences that fold into the pre-designed structures to have targeted functionality is of great importance in various biological applications, including pharmaceuticals, biofuels, biosensors, and agriculture [1, 2, 3, 4, 5]. Mutagenesis aided with high-throughput experiments and physics-based computational methods have been proposed to address this problem in the past [6, 7].

Recently, a new paradigm employing a deep generative model conditioning on three-dimensional structures has gained attention, and its results have been encouraging. These models typically represent a given fixed protein backbone as a 3D graph and leverage advanced generation techniques such as auto-regressive, non-auto-regressive, or adapter-based methods to translate fixed protein backbones into corresponding sequences that can achieve specific biological purposes [8]. For example, Daurass et al. [9] experimentally proved that the deep inverse folding neural networks(IFNN) generates protein sequences with targeted functionality at a higher success rate than Rosetta, a well-established software widely used in the protein community.

As the folding structure is a consequence of the physics that causes folding, acquiring meaningful physical knowledge of folding structures, called folding energy landscape, directly from data would benefit generative models. The currently proposed DIFM has utilized geometric-related features and geometric neural networks to maximize the likelihood of the given training data, aiming to

---

comprehend the protein world and perform inverse-folding based on the physical understanding. However, the experimentally validated structural data is limited to about 200K [10], and the number of structurally distinct and unique proteins practically used for inverse-folding training is only around 30K [11], which is insufficient to train a model to grasp the nature of complex protein folding systems fully. In addition, considering de novo proteins with significantly different characteristics from those discovered thus far, the data scarcity problem worsens more.

In this study, we present DNDesign, a DeNoising-enhanced protein DESIGN framework, which maximizes the model's physical understanding directly from training data. First, we prove that denoising training directly on a three-dimensional backbone structure is equivalent to the direction of energy minimization, finally enabling the models to learn the folding energy landscape. Then, we suggest a novel folding physics learning plug-in module (FPLM), which can be integrated into existing inverse-folding neural networks. First, IFNN takes the original structure, while FPLM inputs the perturbed structures obtained by adding noises to the original structure. Then, FPLM learns the folding energy landscape by moving the perturbed structure to the energetically stable state. To effectively transfer the folding knowledge to IFNN, FPLM employs five novel operations: (1) force feature initialization, (2) force-node attention, (3) force-edge attention, (4) global force attention, and (5) force biasing to transfer the physical inductive bias to the entire network. Notably, the force biasing operation allows the sequence decoder to conduct sampling while considering folding physics. Finally, DNDesign enables IFNN to learn four pieces of information, including original structure, intermediate structure, folding physics, and the direction of energy minimization using given data, enhancing previous methods which only know the original structure.

To demonstrate the novelty of DNDesign, we conduct a series of protein sequence design tasks. Particularly, to showcase the "easy-to-use" nature and generalization of DNDesign, we apply DNDesign to three representative IFNN settings, including auto-regressive (AR), non-auto-regressive (NAR), and scaled-up settings. Remarkably, DNDesign consistently improves the previous method in all experiments, proving that more physical understanding from denoising training benefits models to conduct successful inverse-folding. In addition to considering sequence consistency estimated using the sequence recovery metric, we evaluate the structure conservation by measuring the potential energy change caused by generated sequences after structure relaxation. Based on this metric, we have demonstrated that DNDesign enables models to generate more energetically favorable sequences. To our knowledge, this is the first work comparing structure conservation using potential energy change for evaluating deep inveres-folding models. In addition, extensive ablation and analysis are provided to help the understanding of DNDesign. Our contributions are as follows:

- We propose DNDesign, which enables the inverse-folding model to capture the deep understanding of folding physics that previous models do not fully exploit.

- We prove how DNDesign learns folding physics directly from data and show that DNDesign improves the state-of-the-art model on various protein sequence design benchmarks in auto-regressive, non-auto-regressive, and scaled-up scenarios.

- We introduce a fixed backbone conservation task based on potential energy change from newly generated sequences. The analysis proves that DNDesign generates energetically favorable sequences, leading to more fine-grained protein design.

## 2 Preliminaries and Related works

### 2.1 Protein representation

A protein is a sequence $P = \{A_1, ..., A_N\}$ where $A_i$ is a residue of 20 types of amino acids. Each amino acid is composed of backbone atoms including $C, N, C_\alpha, O$ and side chain $R_i$ consisting of atoms, which determine the property of the amino acid, and each atom has coordinate $\mathbf{x} \in \mathbb{R}^3$. Following ingraham et al [12], each residue has geometry derived from backbone atoms, called local coordinate system $g_i = [b_i, n_i, b_i \times n_i]$, where,

$$u_i = \frac{\mathbf{x}_{C_{\alpha i}} - \mathbf{x}_{N_i}}{\|\mathbf{x}_{C_{\alpha i}} - \mathbf{x}_{N_i}\|}, v_i = \frac{\mathbf{x}_{C_i} - \mathbf{x}_{C_{\alpha i}}}{\|\mathbf{x}_{C_i} - \mathbf{x}_{C_{\alpha i}}\|}, b_i = \frac{u_i - v_i}{\|u_i - v_i\|}. \tag{1}$$

Because each side-chain corresponds to each amino acid residue $s$, a protein can be represented as a sequence of residue and geometry pairs, $P = \{(s_i, g_1), ..., (s_N, g_N)\}$ as depicted in Figure 1.

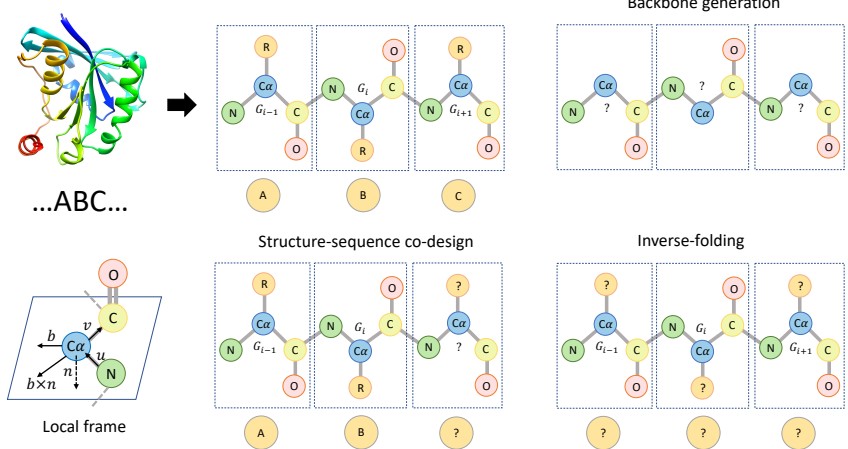

Figure 1: Overview of protein representation and types of structure-based protein design.

## 2.2 Structure-based Protein Design

Because we can reconstruct side-chain $R$ using both backbone $G$ and residues $S$, structure-based protein design only focuses on the generation of backbone $G$ and residues $S$. Neural structure-based protein design can be categorized into three tasks, (1) backbone generation, (2) structure-sequence co-design, and (3) inverse-folding depending on design scenarios as illustrated in Figure 1.

**Backbone generation** Backbone generation can be formulated as an end-to-end structure-to-structure generation as $f_\theta : G \to G$. We train parameterized encoder-decoder neural networks $f_\theta$ to sample diverse and plausible backbone structures. We note that this approach requires a model which can assign amino acids are required to obtain complete proteins [13].

**Structure-sequence co-design** The structure-sequence co-design approach generates sequences and backbone structures simultaneously as $f_\theta : (G, S) \to (G, S)$. By generating sequences autoregressively and refining predicted structures iteratively, this co-design approach enables the adaptive generation of optimized sequences compatible with the flexible backbone. This approach is suitable for scenarios such as antibody design, where global structure tends to be changed depending on the designed antibody [14].

**Inverse-folding** Inverse-folding is an approach that aims to "caption" or "translate" a provided backbone structure into potential protein sequences as $f_\theta : G \to S$. Inverse-folding becomes particularly valuable for protein functional design problems, such as ligand binding site, enzyme, and binder design, which require fine-grained control over side-chain combinations. Since there exist numerous sequences that can fold into a specific backbone conformation [15], inverse-folding approach enables protein engineers to identify diverse and promising protein sequences with optimized functional properties [1, 2, 3, 4, 5].

## 2.3 Importance of physical understanding for structure-based protein design

The folding structure of proteins is governed by the physics that determines the folding process. Therefore, a higher level of understanding of physics enables a model to perform well in structure-related tasks. For instance, AlphaFold2 [16] achieved nearly experimental-level accuracy in structure prediction and has revolutionized protein fields since its publication. Recently, [17] proved that the success of AlphaFold2 [18] is attributed to the data-driven understanding of the folding energy landscape. Additionally, RFDiffusion [13] demonstrated that physically plausible and synthesizable structures were created when generative models were trained upon pre-trained protein structures model, especially, RoseTTAFold [18] that comprehend the mechanism of folding. On the other hand, RefineDiff [19] showed that physics learning enhances protein representation learning for various downstream tasks.

## 3 Related works

### 3.1 Deep generative protein design

Deep generative models for protein design can be classified into four categories: sequence-to-sequence, structure-to-structure, structure-to-sequence, and structure-sequence co-design. In the sequence-to-sequence category, several notable works, such as RITA [20], DARK [21], Prot-GPT2 [22], and ProGen2 [23] have applied language modeling techniques to analyze large protein sequence databases and successfully generated novel protein sequences. Moving on to the structure-to-structure category, deep generative models, such as variational autoencoders [24], generative adversarial networks [25], or diffusion models [26, 27], have been applied to this problem and shown promising results for generating diverse and structurally plausible protein backbones [28, 29, 13, 30, 31, 32]. Third, in the structure-to-sequence category, deep generative inverse-folding models, such as Graph-Trans, GVP, ProteinMPNN, PiFold [12, 33, 34, 35, 36, 9] have been proposed. These models predict or generate protein sequences corresponding to a given protein backbone structure, allowing for the design of functional proteins with specific structural constraints. Last, DiffAb [37] and RefineGNN [38] generate structure and sequence simultaneously through iterative refinement. Our proposed DNDesign falls within the category of deep generative inverse-folding models. DNDesign differentiates itself from previous approaches by utilizing denoising to strengthen the understanding of models.

### 3.2 Denoising training for chemical and biology system modeling

In order to effectively model chemical and biological systems, acquiring meaningful physical knowledge directly from data plays a crucial role in successful representation learning and downstream tasks. Over the past decade, numerous studies have proposed novel deep learning architectures and training methods that demonstrate remarkable performance in approximating interactions and quantum properties [39, 40, 41, 36, 42]. Recently, several works have theoretically established the equivalence between denoising training on biological molecule data and learning the underlying physical force field [43, 44, 19]. Building upon this finding, Godwin et al. [45] successfully addressed the over-smoothing challenge in graph neural networks (GNNs) by employing denoising training on three-dimensional coordinates as an auxiliary loss. Their approach achieved state-of-the-art results on various quantum chemistry downstream tasks. Similarly, [43] utilized denoising as a pre-training objective on the 3M molecular dataset and demonstrated that denoising-based learning produces high-quality representations based on a data-driven force field, effectively solving different molecular benchmarks. In this work, we adopt a similar denoising-based approach to that of [45, 19]. Our work is similar to [19], which utilized denoising to learn folding physics, but they focused on representation learning. Otherwise, our work first extend denoising to protein sequence generation. Despite this difference, the inductive bias gained from DNDesign is identical to that of the two aforementioned papers.

## 4 Methods

### 4.1 Overview

DEDesign integrates conventional inverse-folding networks (IFNN) with denoising networks (FPLM). In this section, we begin by elucidating the process of protein featurization. Subsequently, Section 4.3 outlines the model architecture and describes the feature update operation in IFNN. Next, section 4.4 highlights the correlation between the denoising process and the acquisition of folding physics knowledge. Finally, Section 4.5 describes the five operations utilized to transfer data-driven folding physics across the entire network.

### 4.2 Featurization

In this work, we regard a protein as a k-NN 3D graph $G(V, E, X)$, where $V$, $E$, $X$ are node features, edge features, and xyz coordinates, respectively. As suggested in [12], node and edge features are constructed considering pairwise distances, angle, torsion, and directions using the four backbone atoms of each node. Each node and edge feature is input to a single fully-connected layer and converted into an embedding vector, $h^0, e^0$, respectively. More details are illustrated in Appendix A.

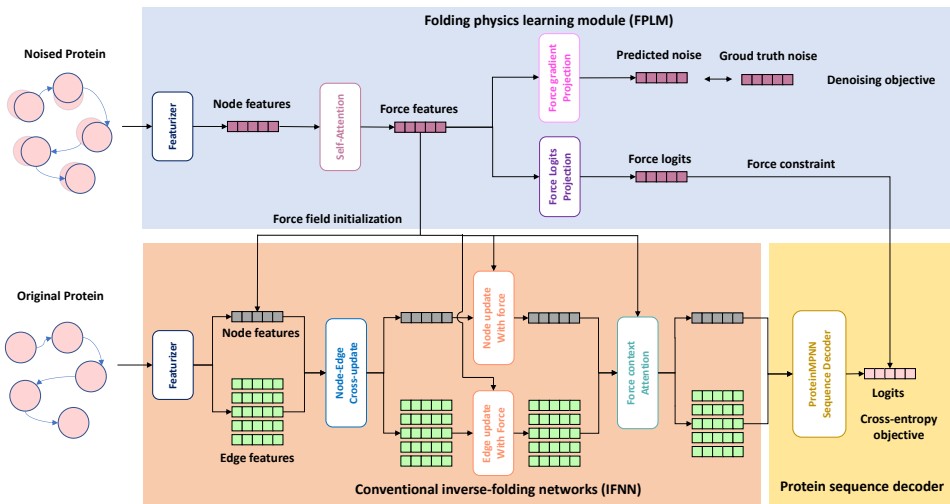

Figure 2: Overview of the proposed DNDesign.

## 4.3 Conventional inverse-folding model modeling

In this subsection, we will briefly describe the IFNN used in this work. As shown in Figure 2, we adopt a state-of-the-art inverse-folding encoder, called PiGNN proposed by [46]. We first update node and edge features in order based on k-NN neighbors. And, the node features $h_i^l$ and edge features $e_i^l$ are obtained using a featurizer and updated using a simplified graph transformer [47]. After computing the node value using the edge features and neighbor node features, we multiply the softmax values of the attention weights obtained from $h_i^l$ and $e_i^l$ to the node value and add the weights to the node features. Finally, we update the $h_i^l$ to obtain the next layer node features $h_i^l$ using a learnable weight function with normalization and residual connection. Edge features $e_{ji}^l$ are calculated by passing concatenation of node feature $h_i$, neighbor node feature $h_j$, and edge feature $e_{i,j}$ to the MLP $\psi^e$.

We use two types of inverse folding decoders to predict logits: (1) auto-regressive (AR) and (2) non-auto-regressive (NAR). For AR, we employ the sequence decoder used in GraphTrans [12] and utilize random decoding introduced by ProteinMPNN [9]. For NAR, we use a linear layer as [33]. The decoder inputs node feature $h_j$ and outputs logits. The whole IFNN is trained by minimizing the negative log-likelihood $\mathcal{L}_{AR}$ of the data.

## 4.4 Folding physics learning via denoising

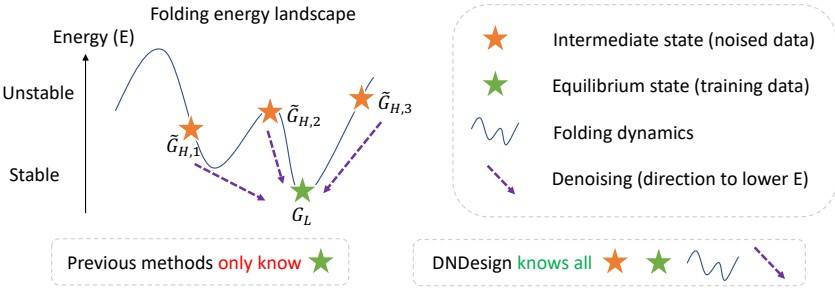

Figure 3: Illustration of Energy landscape of protein folding. DNDesign enable inverse-folding model to capture all information of folding physics.

**Folding energy landscape** The protein structure $G = \{g_1, ..., g_N\}$ that a protein P with sequence $S = \{s_1, ..., s_N\}$ can have is diverse, and each structure corresponds to a specific potential energy $E$

based on the folding physics, represented as the folding energy landscape. Considering the dynamics that structure prefers energetically stable states, each protein structure $G$ used in training data can be regarded as the most stable, i.e., energetically lowest structure $G_L$ among the various folding states that each sequence can possess.

Figure 3 illustrates four crucial pieces of information contained within the energy landscape: (1) stable or low energy state $G_L$, (2) unstable or high energy state $\tilde{G}_H$, (3) folding physics, i.g, energy potential, and (4) the direction or gradient $\nabla G$ that transforms the unstable state $\tilde{G}_H$ to the stable state $G_L$.

**Perturbation on backbone structure**    Firstly, since we only have $G_L$ in the training data, we obtain $\tilde{G}_H$ by adding Gaussian noise to $G_L$. $G$ is a sequence of local frames $g$, where each $g$ consists of a $C_\alpha$ coordinate vector, $\mathbf{x}$, and an $SO3$ vector, $\mathbf{r}$. To account for this nature, we follow the forward noising procedure used in [48, 37]. For $x$, we use random three-dimensional Gaussian noise, which can be described as follows:

$$q\left(\mathbf{x}_j^T \mid \mathbf{x}_j^{T-1}\right) = \mathcal{N}\left(\mathbf{x}_j^T \mid \sqrt{1 - \beta_{pos}^T} \cdot \mathbf{x}_j^{T-1}, \beta_{pos}^T \mathbf{I}\right). \tag{2}$$

$$q\left(\mathbf{x}_j^T \mid \mathbf{x}_j^0\right) = \mathcal{N}\left(\mathbf{x}_j^T \mid \sqrt{\bar{\alpha}_{pos}^0} \cdot \mathbf{x}_j^0, (1 - \bar{\alpha}_{pos}^0)\mathbf{I}\right). \tag{3}$$

where N denotes a Gaussian distribution.

For perturbing $r$, we use an isotropic Gaussian distribution with a mean rotation scalar variance parameter in the SO3 space [48, 49, 50] as follow:

$$q\left(\mathbf{r}_j^T \mid \mathbf{r}_j^0\right) = \mathcal{IG}_{SO(3)}\left(\mathbf{r}_j^T \mid \lambda(\sqrt{\bar{\alpha}_{ori}^T}, \mathbf{r}_j^0), 1 - \bar{\alpha}_{ori}^T\right) \tag{4}$$

, where $\lambda$ is a modification of the rotation matrix by scaling its rotation angle with the rotation axis fixed [51].

**Denoising training**    Based on the remarkable reparameterization technique proposed by Ho et al. [26] and definition of training objective [37], we can optimize denoising objectives for the transition vector $t$ and $SO3$ vector $r$ as follows:

$$L_{pos}^t = \mathbb{E}\left[\frac{1}{M}\sum_j \|\epsilon_j - \psi(\tilde{G}_H, t)\|^2\right], \quad L_{ori}^t = \mathbb{E}\left[\frac{1}{M}\sum_j \|(r_j^0)^{\mathbf{T}}\psi(\tilde{G}_H, t) - \mathbf{I}\|_F^2\right] \tag{5}$$

, where $\psi$ is a neural networks that predicts the perturbation on $t$ and $r$. $L_{pos}^t$ is mean squared error between added Gaussian noise $\epsilon_j$ and predicted noises $\psi(\tilde{G}_H, t)$ and $L_{rot}^t$ minimizes the discrepancy calculated by the inner product between the real and predicted orientation $\psi(\tilde{G}_H, t)$.

**Learning folding physics through denoising learning**    From a statistical perspective, denoising training can be seen as learning the Boltzmann distribution $p_{physical}(G)$, which represents the probability distribution $p_{physical}(G) \propto \exp(-E(G))$ of the structure's energy $E$ using a given protein structure $G$ as a random quantity. Based on the definition, the derivative $\nabla_{\mathbf{x}} \log p_{physical}$ of the folding energy potential $E(G)$ corresponds to the force $-\nabla_{\mathbf{x}} E(\mathbf{x})$ acting on the backbone atoms, directing them towards energetically stable directions as follows:

$$\nabla_{\mathbf{x}} \log p_{physical} = -\nabla_{\mathbf{x}} E(\mathbf{x}) \tag{6}$$

Since $p_{physical}$ is unavailable in practice, we approximate it using data. Following [52], we can compute the log-likelihood of $p_0$ using

$$\log p_0(X(0)) = \log p_T(x(T)) + \int_0^T \nabla \cdot \tilde{f}_\sigma(x(t), t)dt. \tag{7}$$

So, we can approximate $p$ by approximating $\tilde{f}$. To do that, we use Gaussian noise $q_\sigma(\tilde{G}_H^T \mid G_L) = \mathcal{N}(\tilde{G}_H^T; G_L, \sigma^2 \mathcal{I}_{3N})$ as used in other works [27, 26], then, we finally can match the score $\nabla_{\mathbf{x}} \log p_{physical}$ by learning neural networks $\theta(\tilde{G}_H^T, T)$ that predict $\nabla_{\tilde{\mathbf{x}}} \log q_\sigma(\tilde{G}_H^T \mid G_L)$.

$$E_{q_\sigma(\tilde{G}_H^T \mid G_L)}\left[\|\theta(\tilde{G}_H^T, T) - \nabla_{\tilde{\mathbf{x}}} \log q_\sigma(\tilde{G}_H^T \mid G_L)\|^2\right] \tag{8}$$

This score-matching is the same as what we use in the DNDesign framework. So, we can conclude that denoising $\tilde{G}_H$, which has become a higher energy state due to noise, towards $G_L$ is equivalent to learning the folding dynamics.

In summary, in DNDesign, we employ denoising training to learn the energy potential that determines the folding state, i.e., the folding energy landscape. Finally, unlike all previous methods that only train for (1), DNDesign allows for the training of (1), (2), (3), and (4) simultaneously, maximizing the model's physical understanding within the given data.

## 4.5 Physics understanding transfer

This section describes FPLM and the five operations to transfer the folding physics inductive bias of DENN to IFNN.

**Feature embedding** We first extract geometric features of perturbed structure and update the features using orientation-aware roto-translation invariant network networks used in [37]. We call the features force features and use the features in the following five operations.

**Node initialization with force** We add force features to the initial node features so that all node and edge features update in IFNN can be conducted using folding physics understanding.

$$h_i^0 = h_i^0 + f_i^{l+1} \tag{9}$$

**Node update with force** For node updates using force features, we use an attention module that combines two operations: self-attention and cross-attention. The two attentions are based on multi-headed full self-attention [53]. First, self-attention on node features is conducted. Then, cross-attention is performed between the self-attended node features and the force features. At this time, the query and key are force features, and through this, each node strongly interacts with the force state of all other nodes.

**Edge update with force** Edge updates with force features are performed similarly to edge updates with node features in IFNN. After concatenating the force feature $f_i$, the force feature of the neighbor $j$ node $f_j$, and the edge features $e_{ij}$ corresponding to the $ij$ pair, edge feature $e_{ji}^l$ is obtained by projecting the concatenated features into the learnable MLP $\psi^f$.

$$e_{ji}^l = \psi^f(f_j^{l+1}\|e_{ji,1}^l\|f_i^{l+1}) \tag{10}$$

**Global force context attention** The global force state summing all local forces is an essential context for protein sequences. In the above two modules, node and edge updates are conducted using local forces. As the last module, we update node and edge features using the global force state features. The total global force state is simply calculated by taking the average of all force state features. To reduce computational costs, we apply an element-wise product using gated attention combined with a sigmoid function to each node and edge. By doing so, the global force information is transferred to all nodes and edges that comprise a protein.

$$f_i = Mean(\{f_k^{l+1}\}_{k\in\mathcal{B}_i}) \tag{11}$$

$$h_i^{l+1} = h_i^l \odot \sigma(\phi^n(f_i)) \tag{12}$$

$$e_{ji}^{l+1} = e_{ji}^l \odot \sigma(\phi^e(f_i)) \tag{13}$$

**Sequence sampling with end-to-end physics constraint** We obtained logits using force features through linear layer and add the logits to the logits of IFNN as follow:

$$l = \alpha l_s + (1-\alpha)l_f \tag{14}$$

The ratio of force to decoder logits is empirically determined. In this work, we use 0.2 of $\alpha$. By different ratios, we can generate diverse sequences by conditioning on physics features. This adds an additional sampling parameter to inverse-folding sampling, which only resort to probability-based sequence sampling previously.

Table 1: Protein sequence design comparison on CATH 4.2 in both AR and NAR settings. † indicates scores copied from [33], and ‡ indicates the newly calculated scores in our setting.

| Model | Type | Perplexity ↓ | | | Recovery % ↑ | | |
|-------|------|-------|-------------|-----|-------|-------------|-----|
| | | Short | Single-chain | All | Short | Single-chain | All |
| StructGNN† | AR | 8.29 | 8.74 | 6.40 | 29.44 | 28.26 | 35.91 |
| GraphTrans† | | 8.39 | 8.83 | 6.63 | 28.14 | 28.46 | 35.82 |
| GVP† | | 7.23 | 7.84 | 5.36 | 30.60 | 28.95 | 39.47 |
| ProteinMPNN† | | 6.21 | 6.68 | 4.61 | 36.35 | 34.43 | 45.96 |
| PiFold‡ | | 6.31±0.03 | 6.73±0.08 | 4.63±0.01 | 38.50±0.56 | 36.31±0.54 | 48.91±0.28 |
| DNDesign-PiFold‡ | | **5.70**±0.09 | **6.03**±0.08 | **4.49**±0.02 | **39.09**±0.46 | **36.83**±0.49 | **49.88**±0.29 |
| PiFold‡ | NAR | 6.75±0.03 | 7.21±0.10 | 5.05±0.04 | 39.93±0.10 | 37.88±0.52 | 49.49±0.16 |
| DNDesign-PiFold‡ | | **6.72**±0.17 | **7.07**±0.25 | **4.96**±0.04 | **40.18**±0.74 | **38.65**±1.46 | **49.93**±0.42 |

Table 2: Protein sequence design comparison on CATH 4.3 in the scaled-up setting. † indicates scores copied from [8], and ‡ indicates the newly calculated scores in our setting.

| Model | Type | Perplexity | Recovery % ↑ |
|-------|------|-----------|--------------|
| | | All | All |
| GVP-GNN† | AR | 6.06 | 38.20 |
| GVP-Large† | AR | 4.08 | 50.80 |
| GVP-transformer-Large† | AR | 4.01 | 51.60 |
| PiFold‡ | AR | 3.97±0.01 | 52.06±0.08 |
| DNDesign-PiFold‡ | AR | **3.80**±0.01 | **53.75**±0.25 |

## 5 Experiments

In this section, we compare FFDesign with the state-of-the-art deep generative inverse-folding models in three scenarios, including single-chain, multi-chain, and real-world datasets.

### 5.1 Experiment Setting

**Implementation details**   We choose PiFold as IFNN and train PiFold and DNDesign-PiFold in AR, NAR, and scaled-up settings. Models are trained up to 100 in AR and NAR settings, and we set 150 epochs for the scaled-up scenario. All models are trained on 1 NVIDIA A100s with the Adam optimizer [54]. The batch size contains 6000 tokens, and the learning rate is set to 0.001 and decayed with OneCycle scheduler. More details are provided in the appendix. For reliable experiments, all results are obtained using three seeds.

**Baselines**   We employ various graph-based inverse-folding models as baselines, including Struct-GNN, StructTrans [12], GCA [55], GVP [34], GVP-large [8], GVP-transformer [8], AlphaDesign [47], ESM-IF [8], ProteinMPNN [9], and PiFold [33].

### 5.2 Main Results

**Single-chain sequence design**   CATH [11] is a widely used protein dataset to evaluate inverse-folding models on single-chain sequence design tasks. For a fair comparison, we adopt the version of CATH4.2 as used in GraphTrans and GVP, PiFold. In CATH 4.2, 18024, 608, and 1120 proteins are used for training, validation, and testing, respectively. In the standard setting for this task, models are trained using training data and evaluated on three sets from the test set; short-chain, single-chain, and all-set, with perplexity and median recovery scores. Sequence recovery is a widely used metric for inverse-folding and measures how many residues of sampled sequences match that of the ground truth sequence at each position. The results on CATH 4.2 are shown in Table 1. Under similar conditions, the proposed DNDesign consistently improves the previous SOTA method on both perplexity and recovery in both auto-regressive and non-auto-regressive settings.

**Scaling-up**   [8] proved that additional structure data predicted using AlphaFold2 gives remarkable improvement for sequence design. Likewise, we prepare ∼12M predicted structure data of Uniref50 [56] from AlphaFold2 database and train both PiFold and DNDesign-PiFold models using CATH 4.3 + Uniref50. Interestingly, the improvement from denoising becomes more evident in a scaled setting, as shown in Table 2. These results indicate that even with scaling up by adding

millions of structures, a model still accesses only the given stable structures. However, models trained using DNDesign can fully leverage all four pieces of information in folding physics, resulting in improved performance.

**Other benchmarks**   We also conduct sequence design on multi-chain and real-world datasets, TS50, TS500 [57], and Ollikainen [58] datasets. As shown in Appendix A, performance gains from denoising still appear, proving the importance of folding physics understanding for protein inverse-folding design. Detailed results are provided in the appendix.

## 5.3   Analysis

In this section, we provide analyses to understand DNDesign more deeply. This section includes fixed backbone conservation using potential energy and an ablation study about five proposed operations. Additional analyses, such as core-surface analysis and other ablation studies, are provided appendix.

**Fixed backbone conservation study**   Sequence recovery is suitable for measuring sequence consistency, but it cannot fully describe the potential energy behavior. In fixed backbone design, the ultimate goal is to generate sequences with the given structure. Therefore, it is necessary to evaluate whether the generated sequences conserve the fixed backbone structure. Conservation of the given structure implies that the generated sequences pose a structure that do not deviate far from the minimum energy state of the structure. Thus, we can evaluate whether the model generates energetically stable sequences by performing structure relaxation on the sampled proteins, which have corresponding new sequences and given backbone, and measuring the potential energy change. We utilize the Rosetta [59], a well-published computational tool, for structure relaxation. We use 282 structures after filtering structures having residues without coordinates. We generated nine sequences for each structure and performed relaxation for each (structure, sequence) pair. Then, we

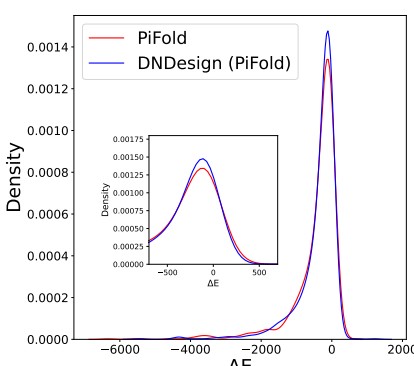

Figure 4: Distribution of potential energy change of structures caused by generated sequences.

computed the change of potential energies. Interestingly, as shown in Figure 4, we observe that PiFold, with an enhanced understanding of folding physics from DNDesign, generates more sequences with potential energy change near zero, meaning complete structure conservation. This clearly demonstrates that our approach, which emphasizes the importance of physics in fixed backbone design and addresses it through denoising, works. To the best of our knowledge, this is the first work comparing fixed backbone conservation using potential energy in fixed backbone design.

**Ablation study**   To understand the effectiveness of each additional force field supervision, we conduct an ablation study, as shown in Appendix B. All components show improvement over baseline, meaning that the folding physics inductive bias is effectively transferred to the entire networks.

## 6   Conclusion

In this study, we have demonstrated that denoising can learn folding physics, and based on this insight, we proposed the DNDesign framework. The proposed framework can easily be integrated into existing SOTA models and has shown performance improvements in various sequence design tasks across multiple settings. Particularly, we evaluated the impact of the acquired energy potential knowledge from the proposed framework by directly assessing its influence on potential energy through the fixed backbone conservation task. This evaluation provides evidence that the model trained with denoising generate energetically favorable sequences. This work sheds light on the significance of learning physics in structure-based protein sequence design, both theoretically and experimentally, and provides energy-based evidence. We hope that our work inspires future research in the structure-based design protein field.

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
