# OpenReview forum: "DNDesign: Denoising is All You Need for Protein Inverse Folding"
_NeurIPS.cc/2023/Conference — Submitted to NeurIPS 2023_

### Official Review · Reviewer_r7Sm · 2023-06-23

**Soundness:** 2 fair
**Presentation:** 2 fair
**Contribution:** 3 good
**Rating:** 4
**Confidence:** 4

**Summary:**

In this work, the authors present DNDesign, a denoising training module atop inverse folding networks (IFNN). The folding physics learning plug-in module (FPLM) is trained following score-matching with noise added to the protein backbone. It also contains five operations, including summation, cross-attention, and gated attention, that integrate the features from FPLM to IFNN. Experimental results show the PiFold with FPLM achieves superior performance on CATH 4.2 and 4.3, when compared to previous IFNNs. Besides, the work introduces a fixed backbone conservation analysis based on potential energy changes to evaluate the performance of IFNNs.

**Strengths:**

1. The protein inverse folding problem that the paper investigates is an emerging domain to apply deep learning techniques.
2. The fixed backbone conservation analysis based on potential energy changes leverages physical prior to evaluate the performance of IFNNs.
3. The idea of utilizing denoising to IFNNs is also connected to physical prior which is expected to boost the performance on inverse folding problems.

**Weaknesses:**

1. The improvement of DNDesign compared to the original PiFold may not be significant.
2. The paper writing needs to be improved. Some notations and technical details are not clearly explained.
3. DNDesign is claimed to be a plug-in for IFNNs. However, it is only tested with PiFold.

Please see "Questions" for more details.

**Questions:**

Major questions:
1. The authors mention proving denoise training is equivalent to learning the direction of energy minimization as a contribution. However, such relationship has been proven in previous works regarding molecules [1,2] and crystals [3]. In this work, the authors apply a similar denoise training strategy to biomolecules. Though the authors cite some of the works. The connection between this paper and previous works should be highlighted.
2. In section 2.2, the authors mention that the side chains can be reconstructed by backbone and residue types, which can be inaccurate. The side chains have degrees of freedom that may not be fully reconstructed from the backbone and residue.
3. In experiments, the authors only apply the proposed FPLM to PiFold and show improvements. How can FPLM be integrated into other IFNNs and how will FPLM affect their performance?
4. Another concern is that the improvement from FPLM may not be significant. In terms of recovery rate, the improvements shown in Table 1 are less than 1%. Besides in Appendix A.1, FPLM shows even worse performance in PP and SR on TS50. And it is worse in SR on TS500 though slightly better in PP. Also in Appendix A.2 for multi-chain sequence design, FPLM shows better performance in 2 out of 4 models though achieving better averaged performance. Further validation of the gain of leveraging the proposed method can be helpful.
5. Also, why apply denoise as an individual plug-in module instead of directly pre-training IFNNs via denoising? Have the authors applied the latter strategy by any chance?
6. In section 5.3, the authors use Rosetta as an oracle in evaluating the potential energy. How accurate is Rosetta in evaluating the energy? The authors are encouraged to discuss the uncertainty.
7. From my perspective, the title may overstate the contribution of the work. The proposed method is a plug-in to existing IFNN but not a general architecture that is "all you need".

Minor questions:
1. In line 35, what is "DIFM"?
2. In line 84, $n_i$ is not defined in the following Eq 1.
3. In Figure 1, the authors may consider denoting the side chains as $R_A$, $R_B$, and $R_C$ as they can be different.
4. In line 198-199, the definition of local frames $g$ is different from line 84. And SO3 vector $\bf r$ is not specified.
In line 233, what is "DENN"?

Reference

[1] Zaidi, S., Schaarschmidt, M., Martens, J., Kim, H., Teh, Y.W., Sanchez-Gonzalez, A., Battaglia, P., Pascanu, R. and Godwin, J., 2022. Pre-training via denoising for molecular property prediction. arXiv preprint arXiv:2206.00133.

[2] Liu, S., Wang, H., Liu, W., Lasenby, J., Guo, H. and Tang, J., 2021. Pre-training molecular graph representation with 3d geometry. arXiv preprint arXiv:2110.07728.

[3] Xie, T., Fu, X., Ganea, O.E., Barzilay, R. and Jaakkola, T., 2021. Crystal diffusion variational autoencoder for periodic material generation. arXiv preprint arXiv:2110.06197.

**Limitations:**

The authors have discussed potential limitations in Appendix B.1.

---

### Official Review · Reviewer_iMTN · 2023-07-02

**Soundness:** 3 good
**Presentation:** 3 good
**Contribution:** 2 fair
**Rating:** 4
**Confidence:** 4

**Summary:**

This work proposes DNDesign, a denoising-enhanced protein fixed backbone design method that effectively captures the protein energy landscape. By integrating denoising training and a plug-in module, DNDesign demonstrates its ability to generate promising protein sequences based on pre-designed structures.
In the inverse folding experiments, the method outperforms all the baseline methods. It also compares the diversity with the baseline, which demonstrates that the method can generate diverse suggestions for a design protein.


update: I keep my score.

**Strengths:**

The authors propose DNDesign, which enables the inverse-folding model to capture the deep understanding of folding physics that previous models do not fully exploit.
They prove how DNDesign learns folding physics directly from data and the method improves the state-of-the-art model on various protein sequence design benchmarks.
The authors further make a fixed backbone conservation task based on potential energy change from newly generated sequences. The analysis proves that DNDesign generates energetically favorable sequences. It is highly likely to work in real wet-lab experiments.

**Weaknesses:**

Figure 2 is a bit confusing. I think the 'noisy protein' and 'protein' should be a protein backbone without side chains?
The energy function and energy based distribution part is a little confusing to me. The distribution (force) is purely learning from the data? Or, the distribution is initialized with some force field, e.g. Rosetta. If the energy is purely learned from the training data, I think the submission should have some experiments to demonstrate that the energy is meaningful.
some typos, e.g. line 158, DEDesign.

**Questions:**

1. In Table 1, the PiFold paper report 51.66% recovery ratio, while this submission report 49.49% accuracy for PiFold. What's the difference between the original PiFold settings and this paper's PiFold settings?
2. In Fixed backbone conservation study, the authors measure the Rosetta energy. I wonder whether the authors can also calculate the AlphaFold scores, which is measured in ProteinMPNN. It can further show the generated protein sequences is of high quality.
3. The denoising step sounds time-consuming. Could the authors provide a comparison of the inference time of PiFold and the proposed method?

---

### Official Review · Reviewer_LQo1 · 2023-07-03

**Soundness:** 3 good
**Presentation:** 2 fair
**Contribution:** 3 good
**Rating:** 5
**Confidence:** 4

**Summary:**

This paper proposed to use denoising diffusion probabilistic model and score-based model to solve the protein inverse folding problem, e.g., generate amino acid sequence given a protein backbone structure. Specifically, this paper used denoising diffusion model to generate unstable protein structure with higher dynamic energy and then used score-based model to learn the physical dynamic forces which drive a protein structure from unstable state to stable one. The learned physical forces are incorporated into a graph-based attention network to predict amino acid sequence. Experiments are conducted on CATH 4.2 and 4.3 datasets and compared with various approaches, showing superior performance (though not by a large margin) over the compared methods.

**Strengths:**

(+) Using score-based model to learn physical dynamics in protein folding is novel, informative and reasonable.

(+) Code has been submitted along with the manuscript submission.



**Weaknesses:**

(-) The paper is not well organized and written, and the logic between each sections/subsections is not obvious and hard to follow.

(-) The texts in Figure 2 are too small to see them clearly with ease.

(-) The experimental results in Table 1 is a little bit marginal compared with PiFold, especial for NAR type.



**Questions:**

Method part:

1. In Figure 3 (left), what the relation among G_{H, 1}, G_{H, 2} and G_{H, 3}? Is G_{H, 2} transformed from G_{H, 1} driven by folding dynamic forces?

2. How to ensure the perturbated backbone structures using Eq.2 - Eq.4 are realistic? I.e., such perturbated structures (or similar ones) do exist in nature?

3. Subsections in Section 4.4 seem to be disconnected. What's the relation between "denoising training" (Line 207, DDPM) and "Learning folding physics through denoising learning" (Line 213, score-based model)? Actually DDPM is a concrete/special case of score-based model, I'm not sure how the authors train the two models (Eq.5 and Eq.8) at the same time in a single DNDesign model.

4. What's the exact meaning of (1) (2) (3) (4) in Line 229?

5. Could the authors provide a more detailed caption for Figure 2 to summarize the workflow of the proposed model?

Experimental part:

6. In Table 4 in Appendix A.4, how to compare the results with and without "Learning folding physics through denoising learning" (Line 213 in the main text)?

7. Is it possible to provide some qualitative results? I.e., given a protein backbone structures visualization, show the predicted sequences and the ground-truth sequences?




**Limitations:**

The authors did not mention any limitations of the proposed method.

---

### Official Review · Reviewer_xi89 · 2023-07-05

**Soundness:** 3 good
**Presentation:** 3 good
**Contribution:** 2 fair
**Rating:** 4
**Confidence:** 4

**Summary:**

This paper combines the denoising pretraining technique with protein inverse folding models, achieving competitive results to baselines. The denoising pretraining has been proven to be effective in molecule and protein representation learning. Therefore, the rediscovery of the phenomenon in protein design is to some extent straightforward.

**Strengths:**

- The paper is clearly written. The presentation is good.
- The methodology is reasonable and convincing.
- The experimental results are positive and support the authors' claims.

**Weaknesses:**

- The innovation seems to be limited. As denoising pretraining has been proven to be effective in molecule [1] and protein [2,3] representation learning, its success in protein design is straightforward.

- The denoising techniques and model design are largely proposed by existing work, which further limits the innovation of this paper

- Compared to the PiFold baseline, the improvement is marginal.

[1] Zaidi, Sheheryar, et al. "Pre-training via denoising for molecular property prediction." arXiv preprint arXiv:2206.00133 (2022).

[2] Huang, Yufei, et al. "Data-Efficient Protein 3D Geometric Pretraining via Refinement of Diffused Protein Structure Decoy." arXiv preprint arXiv:2302.10888 (2023).

[3] Zhang, Zuobai, et al. "Physics-Inspired Protein Encoder Pre-Training via Siamese Sequence-Structure Diffusion Trajectory Prediction." arXiv preprint arXiv:2301.12068 (2023).

**Questions:**

Q1: How do you use the predicted structure data of AlphaFold2? Have you clustered and partitioned these data according to sequence or structure similarity?

Q2: How does the noise scale affect the model? How do the authors adjust the $beta$ and $alpha$ parameters?

Q3: What are the differences of your denoising strategy against previous works[2,3]?

[2] Huang, Yufei, et al. "Data-Efficient Protein 3D Geometric Pretraining via Refinement of Diffused Protein Structure Decoy." arXiv preprint arXiv:2302.10888 (2023).

[3] Zhang, Zuobai, et al. "Physics-Inspired Protein Encoder Pre-Training via Siamese Sequence-Structure Diffusion Trajectory Prediction." arXiv preprint arXiv:2301.12068 (2023).

---

### Decision · Program_Chairs · 2023-09-21

**Decision:**

Reject

**Comment:**

The reviewers felt that the improvements demonstrated were somewhat marginal compared to existing methods like PiFold. Notably, PiFold was not merely slightly more accurate than some of the larger GVP models, but was also significantly more efficient -- a notable considerations when models like ESMFold take (relatively) significant amounts of time per forward pass. The authors did not submit author feedback, so it is difficult to dismiss these concerns.